# Substance use disorders and suicidality in youth: A systematic review and meta-analysis with a focus on the direction of the association

**Charlie Rioux**[1], **Anne-Sophie Huet**[2], **Natalie Castellanos-Ryan**[3,4]*,
**Laurianne Fortier**[3], **Myriam Le Blanc**[5,6], **Stéphanie Hamaoui**[7], **Marie-Claude Geoffroy**[8,9,10],
**Johanne Renaud**[9,10,11], **Jean R. Séguin**[4,5]

**1** Department of Educational Psychology and Leadership, Texas Tech University, Lubbock, Texas, United States of America, **2** Department of Child and Adolescent Psychiatry, University of Ottawa, Ottawa, Ontario, Canada, **3** School of Psychoeducation, Université de Montréal, Montreal, Québec, Canada, **4** CHU Ste-Justine Research Centre, Montreal, Québec, Canada, **5** Department of Psychiatry and Addictology, Université de Montréal, Montreal, Québec, Canada, **6** Institut national de psychiatrie légale Philippe-Pinel, Montreal, Québec, Canada, **7** Department of Psychology, Université du Québec à Montréal, Montreal, Québec, Canada, **8** Department of Educational and Counselling Psychology, McGill University, Montreal, Québec, Canada, **9** Department of Psychiatry, McGill University, Montreal, Québec, Canada, **10** McGill Group for Suicide Studies, Douglas Mental Health University Institute, Montreal, Québec, Canada, **11** Manulife Centre for Breakthroughs in Teen Depression and Suicide Prevention, Montreal, Québec, Canada

* natalie.castellanos.ryan@umontreal.ca

**Data Availability Statement:** All relevant data are within the manuscript and its Supporting Information files.

## Abstract

### Background

Reviews and meta-analyses suggest that substance use and suicidality (i.e., suicidal ideations and attempts) are associated in youth, but the direction of this association remains unclear. Theoretically, the secondary psychiatric disorder hypothesis (SPDH) posits that substance use leads to suicidality, while the secondary substance use disorder hypothesis (SSUDH) posits that suicidality leads to substance use. To clarify these associations, this meta-analysis systematically reviewed studies that examined the prospective associations between SUDs and suicidality in youth (age 25 and younger) and compared results according to the direction of the association.

### Methods

Web of Science, Embase, PsycINFO, PubMed, Medline and ProQuest Dissertations & Theses Global were searched from inception to March 8, 2020, and 55 effect sizes from 23 samples were included and analyzed using a three-level meta-analysis.

### Results

SUDs significantly predicted subsequent suicidality (OR = 2.16, 95%CI 1.57–2.97), suicidality significantly predicted subsequent SUDs (OR = 2.16, 95%CI 1.53–3.04), and these effect sizes did not differ (p = 0.49).

**Funding:** This study was supported by the Fonds Monique Gaumond pour la Recherche en Maladies Affectives through an award to JRS; the Canadian Institutes of Health Research (CIHR; https://cihr-irsc.gc.ca/) through Grant MOP 44072 to JRS, Grant MOP-97910 to JRS, and a fellowship to CR; the Fonds de Recherche du Québec - Santé (FRQS; http://www.frqs.gouv.qc.ca/en/) through a scholarship to CR, a fellowship to CR, a career award to NCR, and a career award to MCG; the American Foundation for Suicide Prevention (https://afsp.org) through a Young Investigator Award to MCG; and the Quebec Network on Suicide, Mood Disorders and Related Disorders (https://reseausuicide.qc.ca) through support to MCG and JR. The funders had no role in study design, data collection and analysis, decision to publish, or preparation of the manuscript.

**Competing interests:** The authors have declared that no competing interests exist.

## Conclusions

Considering that 65% of reviewed studies only examined the SPDH, this review highlights that more attention should be given to the SSUDH, and that studies should examine bidirectional associations between SUDs and suicidality across time. Clinically, because SUDs and suicidality were found to influence each other, results suggest that mental health and SUDs should ideally be detected and treated early, and that co-occurring disorders should be assessed and treated concomitantly.

## Introduction

Adolescence is a vulnerability period for the onset of both substance use (e.g., alcohol and drugs) and suicidal ideations and attempts. Substance use has its onset during adolescence, with heavy episodic drinking and drug use peaking between ages 18–25 [1, 2]. Problematic substance use and substance use disorders (SUDs) in youth are associated with several adverse consequences, including other mental health problems [3–5]. In parallel, suicidality also emerges and peaks during adolescence, and is often a reflection of other mental health problems [6, 7]. According to the World Health Organization [8], suicide is the second leading cause of death among youth ages 15–29. In addition to suicide mortality, other suicidal factors also warrant attention. Suicidal ideation (i.e., thoughts of killing oneself) and non-fatal suicide attempts are predictive of suicidal deaths and are associated with injuries and hospitalization, in addition to having significant emotional repercussions for the adolescent's social network and economic costs for society [9–12].

Studies show that substance use and suicidality are associated in both adolescent and adult populations [13–16], a comorbidity that may be explained developmentally by four hypotheses [17–19]: (1) the *secondary psychiatric disorder hypothesis* (SPDH) posits that SUDs leads to suicidality through an increase in psychological distress and impulsivity, a decrease in effective coping strategies and problem solving, substance-induced depression, decreased quality of social relationships and/or decreased performance at work or school; (2) the *secondary substance use disorder hypothesis* (SSUDH—often referred to as the *self-medication hypothesis*) posits that suicidality leads to SUDs through increased coping motives for substance use (self-medication) and/or using substances to gain acceptance from peers; (3) the *bidirectional hypothesis* posits that there are transactional associations between SUDs and suicidality or that they increase vulnerability for each other; and (4) the *common factor hypothesis* posits that external factors, such as impulse control, broader psychopathology and traumatic life events, are common to both SUDs and suicidality and explain their co-occurrence [17–19]. The present meta-analysis focuses on the SPDH and the SSUDH, although it may have implications for the bidirectional hypothesis since it is a combination of both of the hypotheses of interest.

Although several meta-analyses on the association between substance use and suicidality were recently conducted, they did not allow a comparison of these directional hypotheses. Two meta-analyses with populations of all ages found that alcohol use disorders and SUDs were associated with suicidality [13, 15] and combined findings from cross-sectional and prospective studies, making it difficult to interpret the directionality of effects. Two other meta-analyses were only framed within the SPDH. The first one examined retrospective and prospective studies and found that alcohol and drug use disorders predicted suicide mortality in populations of all ages [16]. The second one examined longitudinal studies (adjusting for suicidality at baseline) and found that cannabis use in adolescence (before 18 years) predicted suicidal ideation and attempt between 18 and 35 years of age [14].

Thus, there is robust evidence regarding the association between substance use and suicidality, but the developmental direction of the association in youth remains unclear. Because co-occurring disorders in youth are related to more severe symptoms, greater treatment challenges and poorer outcomes [20], it is important to understand how SUDs and suicidality are related developmentally. Accordingly, this study will systematically review and analyze studies that examined the prospective associations between SUDs and suicidality in youth (ages 25 and younger), and compare results according to the SPDH and SSUDH.

## Methods

A systematic review and meta-analysis were carried out and is reported in accordance with the Preferred Reporting Items for Systematic Reviews and Meta-Analyses (PRISMA) [21]. As all analyses were based on previously published studies, no ethical approval or informed consent was required. The review protocol was not registered for this study.

### Search strategy

We conducted a search for documents in French, English, Spanish or German in Web of Science^TM, Embase®, PsycINFO®, PubMed®, Medline®, and ProQuest Dissertations & Theses Global^TM, from inception to March 8, 2020. Search terms (provided in S1 Text) for four categories were used: youth, substance use, suicidality, and a prospective/longitudinal design. The search terms were combined using the Boolean operators "OR" within each category, and "AND" between categories. Cross-referencing was used by searching the reference lists of relevant articles. The retrieved titles and abstracts from the search were screened for relevance. The full-text article was evaluated for every abstract that was identified as potentially relevant.

To be included in the review, studies had to meet the following eligibility criteria: (1) quantitative empirical study; (2) text in English, French, Spanish or German; (3) includes data on the relation between SUDs and suicidality; (4) those associations are prospective or longitudinal; and (5) participants are 25 years or younger at follow-up. Exclusion criteria included: (1) experimental study; (2) literature review; (3) only a published abstract is available. In this review, "prospective designs" referred to effects that did not control for the initial levels of the outcome. "Longitudinal designs" controlled for initial levels of the outcome either by controlling for it in the analytical model, coding the outcome measure so it would reflect only new cases, or excluding participants who experienced the outcome at baseline from the analyses.

### Data extraction

Study characteristics (see Results section) and data were extracted into a Microsoft Excel spreadsheet (available upon request). Information extracted from the studies included study identifiers, sample characteristics, suicide measurement, SUD measurement, design, effect sizes (see next paragraph), and methodological quality (see below). Studies were attributed a study number and a sample number identifying studies that analysed the same sample. Effects sizes, variances, and moderating variables were coded in data files for use in the analyses (see below).

We coded effect sizes and standard errors (SE) from each study using the odds ratio (OR). When studies did not include ORs but provided frequency tables, ORs were calculated from these tables. When articles did not provide necessary statistics, the information was requested from authors via electronic mail. Because the distribution of ORs is skewed, the natural log (lnOR) was used for the analyses and transformed back to ORs for reporting the results. Unadjusted ORs were used, wherever possible, to maximize comparability of effect sizes between

studies on both hypotheses and to avoid comparing ORs adjusted for different covariates (see S1 Table for details on covariates in the included effect sizes).

## Analyses

A key assumption in traditional meta-analytic approaches is the independence of effect sizes [22], requiring that all effect sizes be from different samples of participants. However, some studies included more than one effect size (i.e., reporting on more than one SUD or suicidality measure, both prospective and longitudinal associations, several follow-up lengths, and/or age at baseline). Furthermore, multiple studies were published for some samples. Common methods to address this issue include ignoring the dependence of the data, computing an average of dependent effect sizes, or only keeping one effect size per study [22]. However, these methods can bias results and lead to a loss of information. Therefore, a three-level meta-analysis was conducted, which allows the non-independent effect sizes to be clustered [23, 24]. Because there were both multiple effect sizes per study and multiple studies per sample, there were four levels to the data, but current meta-analytic methods are limited to three levels. Thus, the analyses were conducted within samples rather than within studies. Accordingly, the three-level random effects model examined three sources of variance: the sampling variance of the observed effect sizes (level 1), the variance between effect sizes from the same sample (level 2), and the variance between the samples (level 3).

Analyses were first conducted separately for samples examining the SPDH and the SSUDH. Random-effects models were estimated to obtain overall pooled effect sizes and their 95% confidence intervals (CIs). The distribution of effect sizes was examined using tests of heterogeneity with the $Q$ statistic [25]. Significant heterogeneity indicates that differences across effect sizes are likely due to factors other than sampling error, such as different study characteristics. Within-sample and between-sample heterogeneity was quantified using the $I^2$ index; an $I^2$ of 25%, 50%, and 75% respectively reflects a small, medium and large degree of heterogeneity [26]. When there was significant heterogeneity, moderator analyses were conducted to explain variability in effect sizes. The moderators examined included: type of SUD, type of suicidality measure (ideation, attempt; general = measure of ideation and attempt), research design (prospective, longitudinal), type of population (community, at risk, clinical), study quality, follow-up length, proportion of males, proportion of minorities, and year of publication. Because there were few studies examining cannabis use disorder and drug use disorder, they were combined (into a drug use disorder category) for moderation analyses. The moderation analyses in the three-level model are equivalent to a meta-regression analysis, but control for the clustering of effect sizes, and allow examining moderators that vary within and/or between samples. Analyses with cannabis use disorder showed the effect size did not differ significantly from other drug use disorders (see S2 Table). To examine whether the effect size differed according to the direction of association, analyses were then conducted across all studies with the direction of association as a moderator. All analyses were conducted using restricted maximum likelihood estimation and the package metafor [27] in R version 4.0.0 [28] and RStudio version 1.2.5042 [29] on MacOS. The data files and R code are available in S1 Appendix.

## Risk of bias appraisal

Methodological quality of the included studies was assessed with the Newcastle-Ottawa Quality Assessment Scale for cohort studies [30]. The scale assesses the quality of studies based on three categories: participant selection (e.g., representativeness), comparability (i.e., whether the final analysis accounted for important confounding factors), and ascertainment of

outcome (e.g., adequacy of follow-up). Studies can be classified as having good, fair or poor quality (see S2 Text for the full assessment scale).

Publication bias was assessed using Egger's regression test [31, 32], which can be conducted within multilevel random-effects models. A significant test indicates publication bias, or significant funnel plot asymmetry. Funnel plots and trim-and-fill analyses [33] cannot be conducted while taking into consideration the clustering of effect sizes but were used as a complement to Egger's regression test by examining funnel plots with all included effect sizes and conducting trim-and-fill analyses with effect sizes aggregated to one per sample.

## Results

### Included studies and quality assessment

Fig 1 summarizes the results of the literature search. The search yielded 4379 studies after cross-referencing and removing duplicates. 3872 of these studies were excluded based on the titles and abstracts, resulting in 507 full-texts screened. Among those, 27 studies fulfilled the inclusion and exclusion criteria. The required data could not be obtained for two studies [34, 35], resulting in 25 included studies (see Table 1) clustered in 23 samples, with 15 samples examining the SPDH, seven examining the SSUDH, and one examining both hypotheses. Study characteristics are presented in Table 1.

Complete quality assessments for each study can be found in S2 Text. For the SPDH, 7 of the studies had an overall rating of good quality, while 9 had a low rating. Five of those low-quality ratings were due exclusively to the comparability criteria (i.e., the inclusion of important confounders), with the selection and outcome criteria being rated fair or good. Study quality was slightly higher for the SSUDH, with 4 good quality studies, 3 fair quality studies, and 2 poor quality studies, both due to the comparability criteria.

### Secondary psychiatric disorder hypothesis

Analyses for the SPDH included 31 effect sizes nested in 16 samples (see Fig 2 for forest plot of the results). Results showed that SUDs were associated with 2.16 times greater odds (95%CI 1.57–2.97, p < .001) of subsequent suicidal ideations/attempts (95%CI 1.57–2.97, p < .001). There was significant heterogeneity ($Q = 103.29$, df = 30, p < .001) of large magnitude within-samples ($I^2 = 96\%$), but not between-samples ($I^2 = 0\%$). However, moderation analyses (see Table 2) could not explain this heterogeneity, as none of the examined moderators were significant.

### Secondary substance use disorder hypothesis

Analyses for the SSUDH included 24 effect sizes nested in 8 samples (see Fig 3 for forest plot of the results). Results showed that suicidal ideations/attempts were associated with 2.16 times greater odds of subsequent SUDs (95%CI 1.53–3.04, p < .001). There was significant heterogeneity ($Q = 63.56$, df = 23, p < .001) of moderate magnitude within-samples ($I^2 = 45\%$) and of small magnitude between-samples ($I^2 = 21\%$). Moderation analyses (see Table 3) showed that the type of suicidality measure and follow-up length were significant moderators. The effect size was larger for suicidal attempts than for suicidal ideations (t(21) = -2.29, p = 0.03) and general suicidality (t(21) = -2.86, p = 0.01), while effect sizes for suicidal ideations and general suicidality were not significantly different (t(21) = 1.36, p = 0.19; effect sizes in Table 3). A longer follow-up length was associated with a smaller effect size (see Table 3).

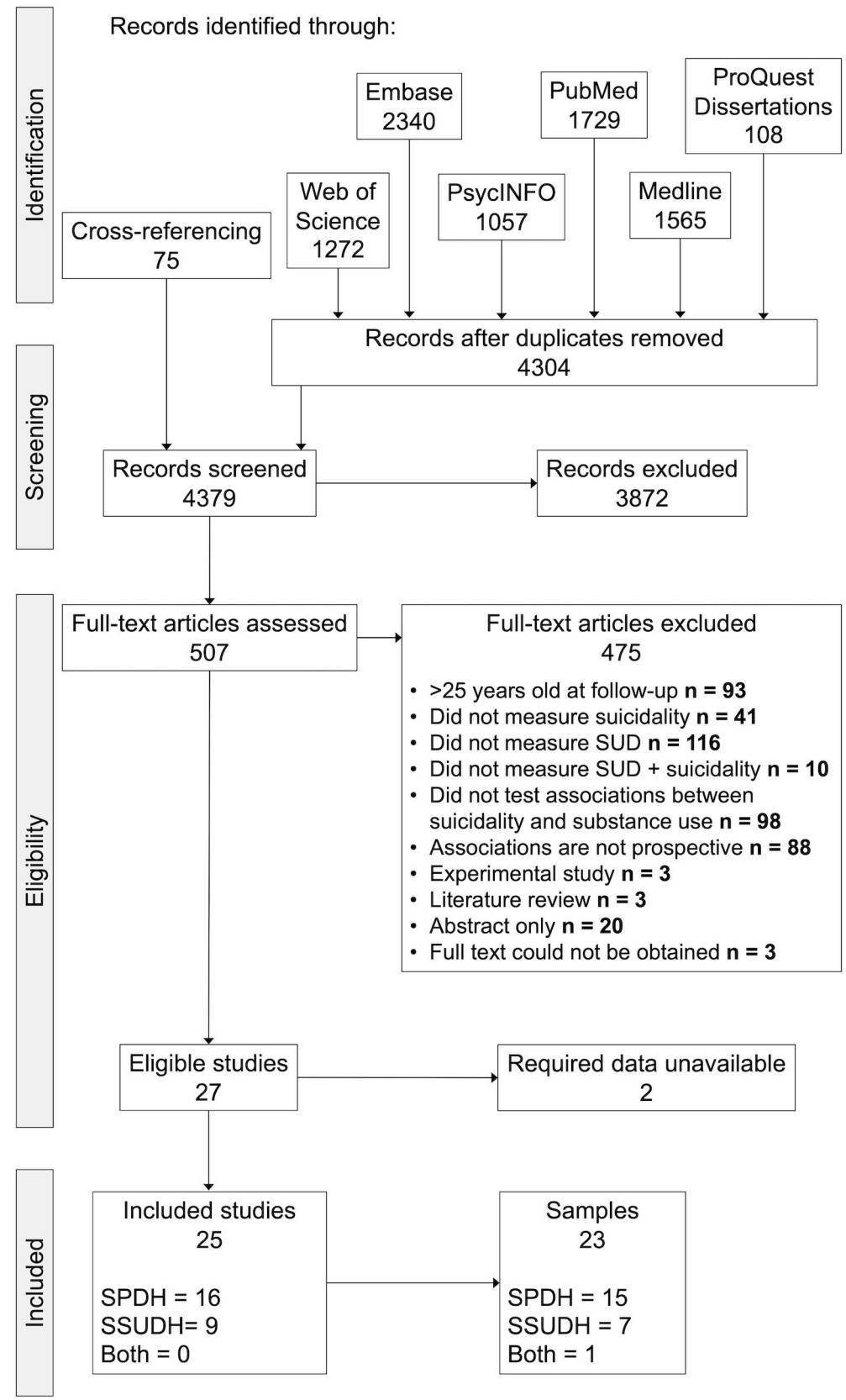

**Fig 1. Flowchart of study selection process.** SPDH = Secondary psychiatric disorder hypothesis (substance use predicts suicidality); SSUDH = Secondary substance use disorder hypothesis (suicidality predicts substance use). See S1 Text for list of full-text articles excluded, with reasons for exclusion.

### Comparison of the hypotheses

Studies for both hypotheses were analyzed together with the direction of association as a moderator. Results showed that the effect size did not differ significantly between both hypotheses $(F(1,53) = 0.49, p = 0.49)$.

### Publication bias

Egger's regression test suggested that there was publication bias for both the SPDH ($p = 0.06$) and the SSUDH ($p = 0.001$). Funnel plots and trim-and-fill analyses (see S1 Fig) suggested that studies with negative effect sizes were missing for the SPDH (i.e., effect sizes missing on the left side of the funnel plot, suggesting the pooled effect size may be larger due to publication bias), while studies with positive effect sizes were missing for the SSUDH (i.e., effect sizes missing on the right side of the funnel plot, suggesting the pooled effect size may be smaller due to publication bias).

### Sensitivity analyses

Sensitivity analyses were conducted to examine whether pooled ORs for each hypothesis were strongly influenced by a single study. Overall (pooled) ORs for each hypothesis were re-estimated, each time leaving out one study. The overall interpretation of the results was not changed by any single study, with ORs all significant and ranging from 1.96 to 2.30 for the SPDH and from 1.96 to 2.39 for the SSUDH (see S3 Table for results of each analysis).

## Discussion

The objective of this systematic review and meta-analysis was to examine the association between SUDs and suicidal ideations/attempts in youth according to the SPDH (i.e., SUD predicts suicidality) and SSUDH (i.e., suicidality predicts SUD). Specifically, prospective and longitudinal studies were reviewed and contrasted according to the direction of association between SUDs and suicidal risk. Results showed that bidirectional associations between substance use and suicidality were significant and did not differ in magnitude according to the direction of association. However, publication bias in opposite directions for the SPDH and SSUDH suggest that the unbiased effect may be slightly larger for the SSUDH.

Although associations were significant and of similar magnitude for both models, there is a clear overrepresentation of the SPDH in the literature. Indeed, 65% of reviewed studies examined this hypothesis exclusively. This is also reflected in previous reviews, with meta-analyses either framing their results within the SPDH [13, 15], or specifically examining this hypothesis [14, 16]. Considering the results of the present meta-analysis, there is a need for studies to give the same attention to the SSUDH, and to favor longitudinal designs which control for pre-substance use psychiatric disorder symptoms when examining the SPDH.

### Direction of association and age groups

Because effects were significant in both directions, results suggest that the bidirectional hypothesis may be the best fit for understanding the association between SUDs and suicidality in youth. Indeed, SUDs and suicidality may be exacerbating each other consistently across development. Another possibility is that effects change direction through developmental

**Table 1. Characteristics of included studies, classified by sample.**

| 1st author, year | Model | n | Mean age T1[a] | Range age T1 | Follow-up (months) | Male (%) | Minority (%)[b] | Country | Population | Design | SUD type | SUD dx system | SUD source | SC type | SC source |
|---|---|---|---|---|---|---|---|---|---|---|---|---|---|---|---|
| **Sample 1: Mexican Adolescent Mental Health Survey** | | | | | | | | | | | | | | | |
| Borges, 2017 [36] | SPDH | 960 | 14.5 | 12–17 | 96 | 51 | NR | MX | Community | L | AUD | DSM-IV | I-A | Ideation | I-A |
| | | 960 | | | 96 | | | | | L | CUD | | | Ideation | |
| | | 960 | | | 96 | | | | | L | DUD | | | Ideation | |
| | | 1041 | | | 96 | | | | | L | CUD | | | Attempt | |
| | | 1041 | | | 96 | | | | | L | DUD | | | Attempt | |
| **Sample 2: EASY Program** | | | | | | | | | | | | | | | |
| Chang, 2015 [37] | SPDH | 700 | 21.2 | 15–25 | 36 | 51 | NR | ROC | C-Psychosis | P | SUD | ICD-10 | CR | Attempt | CR |
| | | 700 | | | 36 | | | | | L | SUD | | | Attempt | |
| **Sample 3: Patterns of Care Study** | | | | | | | | | | | | | | | |
| Chavira, 2010 [38] | SPDH | 831 | 15.2 | 11–18 | 24 | 64 | 67 | US | C-Varied | P | AUD | DSM-IV | I-A | General | I-A |
| | | 818 | | | 24 | | | | | P | DUD | | | General | |
| **Sample 4: National Health Insurance Research Database** | | | | | | | | | | | | | | | |
| Chen, 2019 [39] | SPDH | 275 980 | 9.6 | 4–44 | 168 | 76 | NR | TWN | C-ADHD | P | SUD | ICD-9 | CR | Death | CR |
| **Sample 5: Challenging Times Study** | | | | | | | | | | | | | | | |
| Clarke, 2014 [40] | SPDH | 168 | 13.5 | 12–15 | 96 | NR | NR | IRL | Community | P | CUD | DSM-IV | I-A | Attempt | I-A |
| **Sample 6: Collaborative Study on the Genetics of Alcoholism—Offspring** | | | | | | | | | | | | | | | |
| Conner, 2016 [41] | SPDH | 602 | 14.6 | 12–17 | 25 | 48 | 37 | US | At risk | L | AUD | DSM-IV | I-A | General | I-A |
| | | 602 | 14.6 | | 25 | | | | | L | CUD | | | General | |
| | | 317 | 15.4 | | 25 | | | | | L | AUD | | | General | |
| | | 317 | 15.4 | | 25 | | | | | L | CUD | | | General | |
| **Sample 7: Great Smoky Mountains Study** | | | | | | | | | | | | | | | |
| Copeland, 2017 [42] | SSUDH | 1034 | 11 | 9–13 | 228 | 49 | 31 | US | Community | P | AUD | DSM-IV | I-A | General | I-A I-P |
| | | 1034 | | | | | | | | P | CUD | | | General | |
| **Sample 8** | | | | | | | | | | | | | | | |
| Cox Lippard, 2019 [43] | SPDH | 46 | 18.2 | 13–26 | 3 | 39 | NR | US | C-Mood | L | SUD | DSM-IV | I-A | Attempt | I-A |
| **Sample 9** | | | | | | | | | | | | | | | |
| Dhossche, 2002 [44] | SSUDH | 781 | 14 | 11–18 | 96 | 47 | NR | NL | Community | P | SUD | DSM-IV | I-A | Ideation | SR |
| Herba, 2007 [49] | SSUDH | 1005 | 7.5 | 4–11 | 120 | 49 | NR | NL | Community | P | AUD | DSM-IV | I-A | General | Q-P |
| **Sample 10: Christchurch Health and Development Study** | | | | | | | | | | | | | | | |
| Fergusson, 2005 [45] | SSUDH | 952 | 15 | 15 | 72 | 50 | NR | NZ | Community | P | AUD | DSM-IV | I-A | Ideation | I-A |
| | | 945 | | | 120 | | | | | P | AUD | | | Ideation | |
| | | 887 | | | 120 | | | | | L | AUD | | | Ideation | |
| | | 952 | | | 72 | | | | | P | DUD | | | Ideation | |
| | | 945 | | | 120 | | | | | P | DUD | | | Ideation | |
| | | 887 | | | 120 | | | | | L | DUD | | | Ideation | |
| | | 832 | | | 72 | | | | | P | AUD | | | Attempt | |
| | | 821 | | | 120 | | | | | P | AUD | | | Attempt | |
| | | 887 | | | 120 | | | | | L | AUD | | | Attempt | |
| | | 832 | | | 72 | | | | | P | DUD | | | Attempt | |
| | | 821 | | | 120 | | | | | P | DUD | | | Attempt | |
| | | 887 | | | 120 | | | | | L | DUD | | | Attempt | |

(*Continued*)

**Table 1.** (Continued)

| 1st author, year | Model | n | Mean age T1[a] | Range age T1 | Follow-up (months) | Male (%) | Minority (%)[b] | Country | Population | Design | SUD type | SUD dx system | SUD source | SC type | SC source |
|---|---|---|---|---|---|---|---|---|---|---|---|---|---|---|---|
| **Sample 11** | | | | | | | | | | | | | | | |
| Giaconia, 2001 [46] | SPDH | 365 | 18 | 18 | 36 | 50 | 2 | US | Community | P | DUD | DSM-III-R | I-A | Ideation | SR |
| **Sample 12: COBY Study** | | | | | | | | | | | | | | | |
| Goldstein, 2012 [47] | SPDH | 413 | 12.6 | 7–18 | 65 | 53 | 18 | US | C-Mood | P | SUD | DSM-IV | I-A | Attempt | CR |
| **Sample 13: Avon Longitudinal Study of Parents and Children** | | | | | | | | | | | | | | | |
| Hammerton, 2015 [48] | SPDH | 105 559 | 15 | 15 | 12 | 50 | 3 | UK | Community | P | AUD | DSM-IV | I-A | Ideation | SR |
| Mars, 2014 [54] | SSUDH | 4799 | 16 | 16 | 24 | 41 | 4 | UK | Community | P | AUD | ICD-10 | I-A | Attempt | SR |
| | | | | | | | | | | P | CUD | DSM-IV | | Attempt | |
| **Sample 14: Hawaiian High Schools Health Survey** | | | | | | | | | | | | | | | |
| Hishinuma, 2018 [50] | SPDH | 2083 | 15 | 14–16 | 12 | 46 | 36 | US | Community | P | SUD | NR | SR | Attempt | SR |
| **Sample 15** | | | | | | | | | | | | | | | |
| Iorfino, 2018 [51] | SSUDH | 1143 | 18.8 | 12–30 | 21 | 43 | NR | AUS | C-Varied | P | SUD | DSM-V | CR | Attempt | CR |
| | | | | | | | | | | L | SUD | | | Attempt | |
| **Sample 16: Emergency Department Screening for Teens at Risk for Suicide Study** | | | | | | | | | | | | | | | |
| King, 2019 [52] | SPDH | 2104 | 15.1 | 12–17 | 3 | 37 | 47 | US | Community | P | AUD | ICD-10 | SR | Attempt | I-A I-P |
| **Sample 17: Oregon Adolescent Depression Project** | | | | | | | | | | | | | | | |
| Lewinsohn, 2001 [53] | SPDH | 941 | 16 | 14–18 | NR | 0 | NR | US | Community | P | AUD | DSM-III-R | I-A | Attempt | I-A |
| | | 941 | | | | 100 | | | | P | AUD | | | Attempt | |
| | | 941 | | | | 0 | | | | P | DUD | | | Attempt | |
| | | 941 | | | | 100 | | | | P | DUD | | | Attempt | |
| **Sample 18** | | | | | | | | | | | | | | | |
| Miranda, 2014 [55] | SPDH | 506 | 15.6 | 12–21 | 61 | 39 | 56 | US | Community | L | SUD | DSM-III-R | I-A | Attempt | I-A |
| **Sample 19: Medicaid Database** | | | | | | | | | | | | | | | |
| Olfson, 2018 [56] | SPDH | 32 356 | 17 | 12–24 | 12 | 32 | 38 | US | At risk | P | SUD | ICD-9 | CR | Death | CR |
| **Sample 20** | | | | | | | | | | | | | | | |
| Reinherz, 1995 [57] | SSUDH | 364 | 15 | NR | 36 | 51 | 1 | US | Community | P | AUD | DSM-III-R | I-A | Ideation | SR |
| | | | | | | | | | | P | DUD | | | Ideation | |
| **Sample 21** | | | | | | | | | | | | | | | |
| Skarbo, 2004 [58] | SSUDH | 100 | 15.9 | 10–22 | 84 | 19 | NR | NOR | C-Varied | L | SUD | DSM-IV | I-A | Attempt | CR |
| **Sample 22: Zurich Adolescent Psychology and Psychopathology Study** | | | | | | | | | | | | | | | |
| Steinhausen, 2006 [59] | SSUDH | 98 | 13.6 | 11–17 | 72 | 48 | NR | CH | Community | P | SUD | DSM-IV | I-A | General | SR |
| **Sample 23: Adolescent Depression Study** | | | | | | | | | | | | | | | |
| Tuisku, 2014 [60] | SPDH | 137 | 16.4 | 13–19 | 12 | 18 | NR | FIN | C-Mood | P | AUD | DSM-IV | SR | Attempt | I-A |
| | | 137 | | | 12 | | | | | L | AUD | | | Attempt | |
| | | 137 | | | 97 | | | | | P | AUD | | | Attempt | |
| | | 137 | | | 97 | | | | | L | AUD | | | Attempt | |

AUD = Alcohol use disorder; AUS = Australia; C- = Clinical-; CH = Switzerland; CR = Clinical record; CUD = Cannabis use disorder; DUD = Drug use disorder; FIN = Finland; I-A = Interview-Adolescent; I-P = Interview-Parent; IRL = Ireland; L = Longitudinal; MX = Mexico; NL = Netherlands; NOR = Norway; NR = Not reported; NZ = New-Zealand; P = Prospective; Q-P = Questionnaire-Parent; ROC = China; SC = Suicidality; SPDH = Secondary psychiatric disorder hypothesis; SR = Self-Report; SSUDH = Secondary substance use disorder hypothesis; SUD = Substance use disorder; T1 = Time 1 (baseline); TWN = Taiwan; US = United States.

[a]When only age range was provided, mid-age range was used. When only grade levels in US school system were reported, age was calculated using grade+5.

[b]Minority refers to ethnic minority.

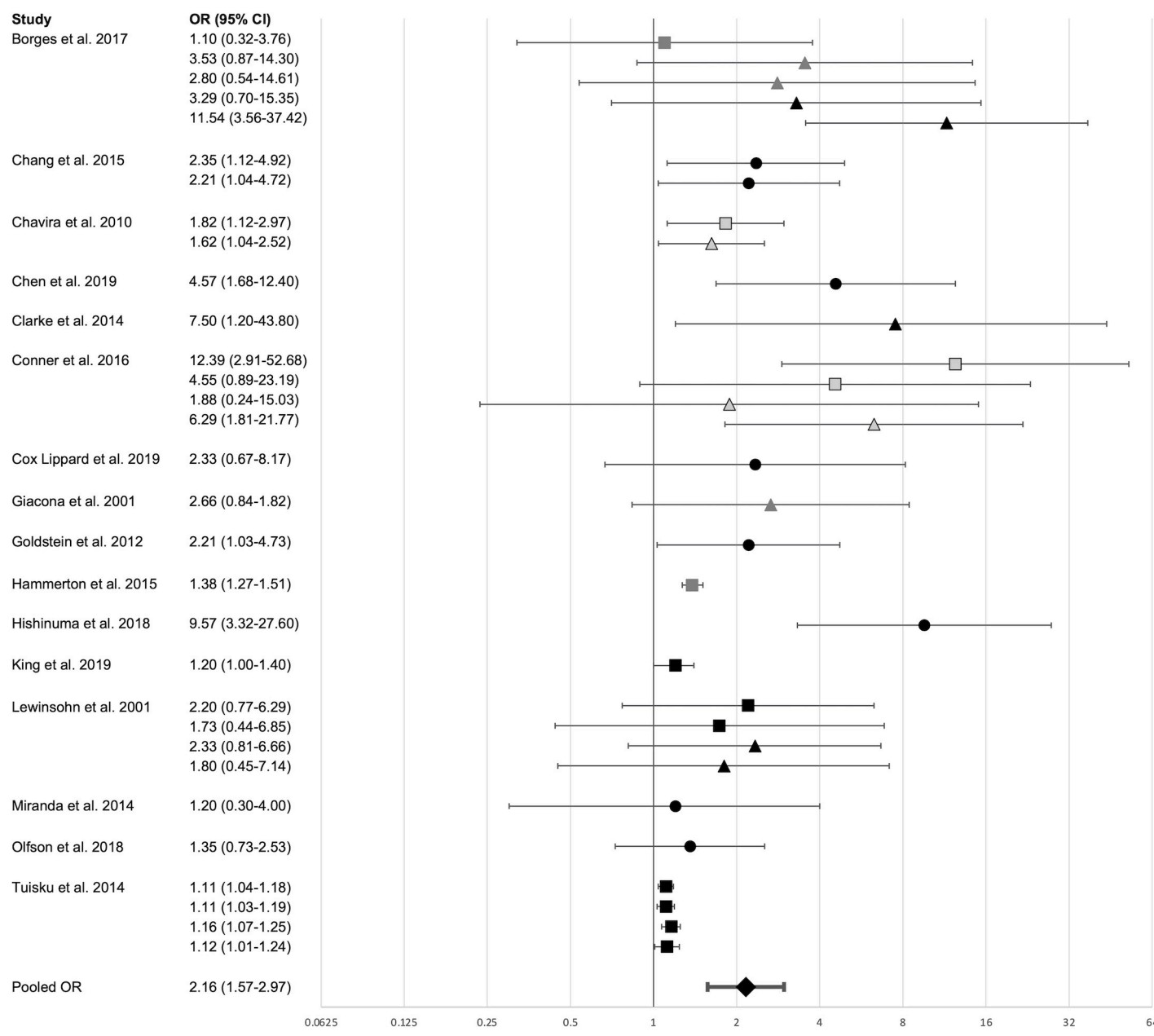

**Fig 2. Secondary psychiatric disorder hypothesis forest plot.** Markers represent effect sizes, bars represent the 95% confidence interval. Pooled OR based on three-level random effects model.

transactions. Unfortunately, the wide variety of average ages at study onset, age ranges and follow-up lengths in the present meta-analysis did not allow examining age as a moderator. Cross-lagged panel studies would be well-suited to clarify these bidirectional changes over time [61].

Although none of the studies included in the meta-analysis on SUDs examined bidirectional effects, two studies tested such a model using substance use frequency and suicidal ideation. One longitudinal study ranging from ages 13 to 16 years of age found that suicidal

**Table 2. Results of the three-level meta-analytic model for the secondary psychiatric disorder hypothesis (SPDH).**

| Variable | #Samples | #ES | n | OR (95% CI) | B (SE) | F(df1, df2) | Q |
|---|---|---|---|---|---|---|---|
| SUD | 16 | 31 | 423 832 | 2.16 (1.57–2.97) | | | 103.29*** |
| Categorical moderators | | | | | | | |
| Type of SUD | | | | | | F(2, 28) = 1.50, p = .24 | 64.26*** |
| General (SUD) | 7 | 8 | 312 084 | 2.49 (1.54–4.04) | | | |
| Alcohol (AUD) | 7 | 12 | 111 134 | 1.71 (1.17–2.48) | | | |
| Drug (DUD) | 6 | 11 | 3 935 | 2.38 (1.50–3.77) | | | |
| Suicidality | | | | | | F(2, 28) = 1.14, p = .33 | 72.72*** |
| General | 2 | 6 | 1 433 | 2.92 (1.16–7.35) | | | |
| Ideation | 3 | 5 | 106 884 | 1.39 (0.66–2.91) | | | |
| Attempt | 12 | 20 | 316 475 | 2.35 (1.55–3.55) | | | |
| Study design | | | | | | F(1, 29) = 0.08, p = .78 | 100.91*** |
| Prospective | 12 | 17 | 421 637 | 2.17 (1.58–2.99) | | | |
| Longitudinal | 6 | 14 | 3 032 | 2.15 (1.55–2.97) | | | |
| Population | | | | | | F(2, 28) = 0.21, p = .81 | 72.08*** |
| Community | 8 | 15 | 112 767 | 2.26 (1.38–3.71) | | | |
| At risk | 2 | 5 | 32 958 | 2.73 (1.07–6.99) | | | |
| Clinical | 6 | 11 | 278 107 | 1.97 (1.14–3.38) | | | |
| Study quality | | | | | | F(1, 29) = 1.64, p = .21 | 89.15*** |
| Poor | 9 | 17 | 140 180 | 2.58 (1.69–3.93) | | | |
| Good | 7 | 14 | 283 652 | 1.74 (1.09–2.77) | | | |
| Continuous moderators | | | | | | | |
| Follow-up length | 15 | 27 | 422 981 | | 0.000 (0.001) | F(1, 25) = 0.89, p = .36 | 99.72*** |
| Proportion of males | 15 | 30 | 423 664 | | 0.003 (0.005) | F(1, 28) = 0.33, p = .57 | 60.11*** |
| Proportion of minorities | 9 | 13 | 144 819 | | -0.002 (0.011) | F(1, 11) = 0.06, p = .81 | 36.54*** |
| Year of publication | 16 | 31 | 423 832 | | 0.010 (0.031) | F(1, 29) = 0.10, p = .76 | 103.02*** |

AUD = Alcohol use disorder, DUD = Drug use disorder, SUD = Substance use disorder

ideation at 14 years predicted alcohol use frequency at 15 years, but alcohol use did not predict suicidal ideation [62]. The other longitudinal study examined cannabis use frequency from 15 to 20 years of age and found that weekly cannabis use at 15 years predicted suicidal ideation at 17 years, but suicidal ideation did not predict cannabis use—and the association found was fully explained by alcohol, tobacco, and other drug use [63]. This raises the possibility that the SSUDH would be most relevant in early adolescence, when suicidal thoughts and behaviors are prevalent, but substance use prevalence only begins to increase, whereas the SPDH would be most relevant later in adolescence, when quantity and frequency of substance use, as well as SUDs, are higher. This should be examined in studies with an age range covering the whole adolescent period. Further, as for now, studies on bidirectionality have focused on substance use frequency and suicidal ideation, however, studies including SUDs and suicidal attempts are needed. Finally, it should be noted that although such analyses can clarify the directionality of effects, they do not demonstrate causality.

## Moderators

Several moderators were examined in the present meta-analysis. Neither models were moderated by the type of SUD (i.e., general SUD, alcohol use disorder, drug use disorder). However, the effects reported for each substance could be considered general substance use effects as

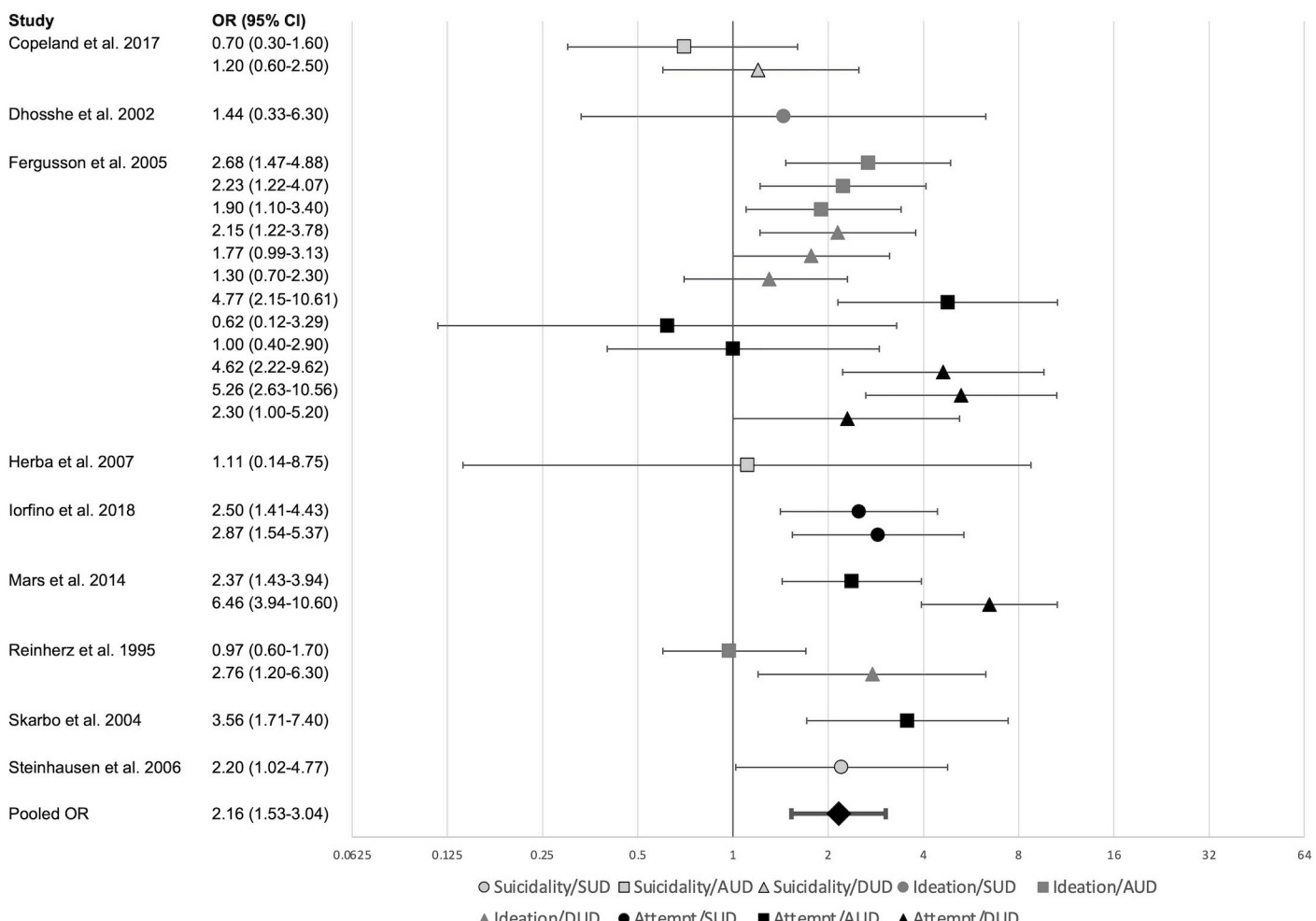

**Fig 3. Secondary substance use disorder hypothesis forest plot.** Markers represent effect sizes, bars represent the 95% confidence interval. Pooled OR based on three-level random effects model.

most studies reviewed did not control for the common variance across substances or examine specific effects. Only one study examined substance specific effects, showing that cannabis use disorder was associated with increased odds of suicide attempt even when taking into account alcohol use [40]. Thus, more studies are needed to clarify general from substance-specific effects, as well as the effects of poly-substance use. The effect size of suicidality predicting SUDs was larger for suicidal attempts than for suicidal ideations for the SSUDH, but not for the SPDH. Furthermore, a longer follow-up length was associated with a smaller effect size for the SSUDH, but not for the SPDH. However, there were longer follow-up lengths for the SSUDH (ranging from 21 to 228 months) than for the SPDH (ranging from 3 to 168 months). Accordingly, these results suggest that the effect of SUDs predicting suicidality is longer lasting than the effect of suicidality predicting SUDs, and/or the interinfluence between SUDs and suicidality gets smaller after several (7+) years. Longitudinal studies examining bidirectional effects between SUDs and suicidality at different intervals could help clarify how long the prediction of one by another lasts.

Significant moderators explained some, but not all, variance in effect sizes for the SSUDH. Large within-sample heterogeneity could not be explained for the SPDH, as no significant

**Table 3. Results of the three-level meta-analytic model for the secondary substance use disorder hypothesis (SSUDH).**

| Variable | #Samples | #ES | n | OR (95% CI) | B (SE) | F(df1, df2) | Q |
|---|---|---|---|---|---|---|---|
| Suicidality | 8 | 24 | 9 495 | 2.16 (1.53–3.04) | | | 63.56*** |
| Categorical moderators | | | | | | | |
| Type of SUD | | | | | | F(2, 21) = 1.29, p = .30 | 57.67*** |
| General (SUD) | 3 | 4 | 2 022 | 2.29 (1.07–4.91) | | | |
| Alcohol (AUD) | 6 | 11 | 8 254 | 1.78 (1.08–2.94) | | | |
| Drug (DUD) | 4 | 9 | 7 149 | 2.58 (1.51–4.42) | | | |
| Suicidality | | | | | | F(2, 21) = 5.05, p = .01** | 40.71* |
| General | 3 | 4 | 3 061 | 1.24 (0.68–2.24) | | | |
| Ideation | 3 | 9 | 2 097 | 1.92 (1.41–2.62) | | | |
| Attempt | 4 | 11 | 6 929 | 3.15 (2.27–4.36) | | | |
| Study design | | | | | | F(1, 22) = 1.66, p = .21 | 61.02*** |
| Prospective | 8 | 19 | 9 495 | 2.26 (1.53–3.31) | | | |
| Longitudinal | 2 | 5 | 2 030 | 1.59 (0.86–2.93) | | | |
| Population | | | | | | F(1,22) = 0.93, p = .35 | 61.83*** |
| Community | 6 | 21 | 8 252 | 1.96 (1.29–2.97) | | | |
| Clinical | 2 | 3 | 1 243 | 2.97 (1.35–6.51) | | | |
| Study quality | | | | | | F(2,21) = 0.53, p = .60 | 51.41*** |
| Poor | 2 | 3 | 462 | 1.70 (0.79–3.73) | | | |
| Fair | 2 | 4 | 5 804 | 2.94 (1.33–6.48) | | | |
| Good | 4 | 17 | 3 229 | 2.10 (1.33–3.33) | | | |
| Continuous moderators | | | | | | | |
| Follow-up length | 8 | 24 | 9 495 | | -0.005 (0.002) | F(1, 22) = 6.59, p = .02** | 50.17*** |
| Proportion of males | 8 | 24 | 9 495 | | -0.030 (0.019) | F(1,22) = 2.52, p = .13 | 54.75*** |
| Proportion of minorities | 3 | 6 | 6 197 | | -0.032 (0.032) | F(1,4) = 0.97, p = .38 | 30.53*** |
| Year of publication | 8 | 24 | 9 495 | | 0.008 (0.024) | F(1,22) = 0.11, p = .75 | 60.31*** |

AUD = Alcohol use disorder, DUD = Drug use disorder, SUD = Substance use disorder

moderators were found. This suggests that some factors that vary within samples but could not be examined in this meta-analysis may moderate the association between SUDs and suicidality and should be investigated in future empirical studies. For example, although the percentage of males in the samples did not moderate results in this meta-analysis, effects may still differ between sexes when examined separately. While odds ratios by sex were not available in enough studies to be examined here, moderation by sex was examined in a few studies. For the SPDH, only one study examined moderation by sex, but SUDs were not associated with later suicide attempts for either sexes [53]. Studies on substance use frequency found that the prediction of suicidality by substance use frequency was significant for both sexes [64–67], and one study found that early onset cannabis and inhalant use (i.e., use before 15 years) were associated with suicidal ideations and attempts at 19 years in girls, but not in boys [68]. For the SSUDH, one study found that suicidal ideation and attempts before 18 years were associated with SUDs between 18 and 25 years for girls, but not for boys [45], while one study found the association between suicidal ideations at 15 years and drug use disorders at 18 years was not moderated by sex [57]. Thus, the moderation of these associations remains unclear and may differ by type of substance use, but results suggest that sex is an important moderator to consider in future studies. Likewise, no studies examined moderation by gender, which should also be considered in future studies, especially considering that measures of gender (i.e., of masculinity and femininity) have been shown to account for differences which may have

otherwise been attributed to sex if gender had not been examined in the prediction of drug use in emerging adulthood [69].

Similarly, ethnic minority status in the samples did not moderate results in this meta-analysis, but these analyses were particularly low powered (9 studies for the SPDH, 3 studies for the SSUDH), and effects may still differ between specific ethnic groups when examined separately. Studies on SUDs did not examine this moderation, which was only tested in one study on substance use frequency and the SPDH hypothesis that found no moderation by ethnicity, with alcohol, cannabis, and other drug use frequencies predicting suicide attempts in Black, Hispanic, and white youth. Importantly, over half of the studies reviewed did not provide any information about the ethnic distribution of their sample. While studies examining the moderation of the association between SUDs and suicidality by ethnicity are needed, all studies should provide these demographics to allow for proper interpretation of the results and their generalizability.

In terms of other moderators, one of the reviewed studies also found that alcohol use disorder interacted with the quality of father-child relationship to predict suicidal ideation [41], and other family factors could be examined in future studies (e.g., parenting style, parental monitoring, parent-child communication [70, 71]). Other correlates of substance use and suicidality that could be moderators of these associations but have not been examined in the studies reviewed notably include peer affiliations [72, 73], personality [71, 74], and sexual minority status [75].

## Limitations

The present review is, to our knowledge, the first meta-analysis comparing the direction of the association between SUDs and suicidality in youth. It has important strengths, including a comprehensive systematic literature search and multilevel meta-analysis, which allowed multiple effect sizes per sample to be included, made it possible to examine some potentially important moderators, and can yield more precise average effect size estimates. Still, some limitations should be noted.

First, despite the multilevel approach, some moderation analyses did not include all studies, and were thus underpowered. Many moderators included levels with fewer than five studies, which is not ideal for ensuring adequate coverage and accurate results [22]. Once more studies on the topic are published, future meta-analyses should conduct more reliable and well-powered moderation analyses. Second, over half of the studies for the SPDH and a fourth of the studies for the SSUDH were of poor quality according to the Newcastle-Ottawa criteria [30]. The lower quality of studies for the SPDH may have affected the robustness of the results comparing the directional hypotheses. However, moderator analyses indicated that results did not significantly differ between studies of good and poor quality. Still, the evaluation of study quality highlighted the need for studies of high quality, especially studies controlling for potential confounders, and studies including other substance use to disentangle substance-specific effects. Further, a majority of studies had attrition rates over 10% and used deletion techniques, and future studies should aim to use appropriate missing data treatments to avoid bias when attrition and missing data are present [76, 77]. Third, although studies from various countries were included, the majority of studies were from the western world and there were no studies from South America and Africa, which may affect the generalizability of the results. One study from South Africa found that alcohol use frequency in 10-18-year-olds did not significantly predict suicidal ideation/attempts one year later [78]. However, more studies from this region, including studies on SUDs, would be needed to properly assess whether associations are similar or different from those found in other countries. Finally, this meta-analysis

only included published studies written in French, Spanish, German, or English, possibly increasing the risk of publication bias in the results. Analyses suggested that publication bias was present, and differential bias between the SPDH and the SSUDH may have impacted the moderation analyses comparing the two hypotheses.

## Clinical implications

Despite its limitations, the results of the present meta-analysis suggest that the associations between SUDs and suicidality are bidirectional. Although caution is warranted in the interpretation of results, and replication of findings in future studies specifically designed to examine bidirectionality is warranted, these findings have potential but clear implications for clinical practice and policy.

First, findings support the importance of detecting and treating SUDs and other mental health problems early to prevent co-morbidity, which is associated with poorer treatment outcomes [20]. In addition to assessing substance use as a potential etiological factor in youth who present with mental health problems, clinicians should also be watchful for the initiation of substance use by their patients, which they might not be as attentive to as reflected by the publication bias towards the SPDH. While a recent meta-analysis suggested that public health programs should target adolescent cannabis use to prevent suicidality [14], the present meta-analysis highlights that these prevention efforts should target all substance use, in addition to also aiming to prevent the exacerbation of substance use by suicidality itself and its root causes. A cross-influence of SUDs with suicidality could notably be prevented by targeting the mechanisms thought to explain their associations, for example by increasing coping and problem-solving skills [17, 18]. Some programs targeting these skills have already been shown to prevent both substance use [79, 80] and suicidality [81, 82], and may also prevent co-morbidity when youth already experience one or the other, which could be examined in future experimental research.

## Conclusions

In conclusion, the present meta-analysis showed that SUDs and suicidality in youth likely influence each other, an effect that was significant for alcohol and drugs, as well as suicidal ideation and attempts. Since the majority of research focuses on the prediction of suicidality by SUDs, this review highlighted that more attention should be given to how suicidality predicts later SUDs, and that research on the transactional relationship between these two constructs is needed, in addition to research on the moderators and mediators of these associations. A better understanding of the developmental and reciprocal association between substance use and suicidality and how these may change across developmental periods will help improve evidence-based prevention and intervention programs by identifying age-appropriate targets.

## Supporting information

**S1 Checklist. PRISMA 2009 checklist.**
(DOC)

**S1 Fig. Funnel plots.**
(PNG)

**S1 Table. Covariates in the included effect sizes.**
(DOCX)

**S2 Table. Analyses with cannabis and other drugs examined separately.**
(DOCX)

**S3 Table. Sensitivity analyses.**
(DOCX)

**S1 Appendix. Data and R code.**
(ZIP)

**S1 Text. Literature search.**
(DOCX)

**S2 Text. Quality assessment.**
(DOCX)

## Acknowledgments

We thank the many authors who contributed supplementary data, which allowed us to include their manuscripts in this review.

## Author Contributions

**Conceptualization:** Charlie Rioux, Anne-Sophie Huet, Natalie Castellanos-Ryan, Myriam Le Blanc, Stéphanie Hamaoui, Marie-Claude Geoffroy, Johanne Renaud, Jean R. Séguin.

**Data curation:** Charlie Rioux, Anne-Sophie Huet, Laurianne Fortier.

**Formal analysis:** Charlie Rioux.

**Methodology:** Charlie Rioux, Anne-Sophie Huet, Natalie Castellanos-Ryan, Jean R. Séguin.

**Project administration:** Jean R. Séguin.

**Supervision:** Charlie Rioux, Natalie Castellanos-Ryan, Marie-Claude Geoffroy, Jean R. Séguin.

**Visualization:** Charlie Rioux.

**Writing – original draft:** Charlie Rioux.

**Writing – review & editing:** Charlie Rioux, Anne-Sophie Huet, Natalie Castellanos-Ryan, Laurianne Fortier, Myriam Le Blanc, Stéphanie Hamaoui, Marie-Claude Geoffroy, Johanne Renaud, Jean R. Séguin.

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
