## [Decision Letter · Decision Letter 0]

18 May 2021

PONE-D-20-38935

Substance use disorders and suicidality in youth: A systematic review and meta-analysis with a focus on the direction of the association

PLOS ONE

Dear Dr. Castellanos-Ryan,

Thank you for submitting your manuscript to PLOS ONE. After careful consideration, we feel that it has merit but does not fully meet PLOS ONE’s publication criteria as it currently stands. Therefore, we invite you to submit a revised version of the manuscript that addresses the points raised during the review process.

The two reviewers suggested several comments, which will improve the integrity of the paper.

We look forward to receiving your revised manuscript.

Kind regards,

Kyoung-Sae Na, M.D.

Academic Editor

PLOS ONE

Journal Requirements:

2. We note that this manuscript is a systematic review or meta-analysis; our author guidelines therefore require that you use PRISMA guidance to help improve reporting quality of this type of study. Please upload copies of the completed PRISMA checklist as Supporting Information with a file name “PRISMA checklist”.

Reviewers' comments:

Reviewer's Responses to Questions

**Comments to the Author**

1. Is the manuscript technically sound, and do the data support the conclusions?

Reviewer #1: Yes

Reviewer #2: Yes

2. Has the statistical analysis been performed appropriately and rigorously? 

Reviewer #1: Yes

Reviewer #2: Yes

3. Have the authors made all data underlying the findings in their manuscript fully available?

Reviewer #1: Yes

Reviewer #2: Yes

4. Is the manuscript presented in an intelligible fashion and written in standard English?

Reviewer #1: Yes

Reviewer #2: Yes

5. Review Comments to the Author

Reviewer #1: Overall, the manuscript by Rioux and colleagues will make an important contribution to the literature. The manuscript has multiple strengths including the following.

1) It examines direction in the relationship between SUD and suicide in the current literature and helps to summarize what is know on this important topic.

2) The study uses a well thought out methodology that that addresses potential biases in analyses. The authors present extensive analyses in detail to support their results.

3)The manuscript is well written and organized and includes discussion of important limitations. It also includes a comprehensive discussion next steps.

However, there are a few minor issues that should be addressed.

1) Overall, the discussion section would be improved if the paragraphs were broken up further. For instance, from page 19 line 291 to page 20 line 328 is one paragraph discussing moderators. This could be broken into several paragraphs. This issue is present throughout the discussion section, subsections (e.g., “limitations”) would help to navigate this discussion.

2) The discussion of substance abuse frequency is not entirely clear. Lines 300-305 discusses this issue. Although the opening sentence indicates discussion of substance use frequency, the later lines use the term “substance use” instead. It is not clear that whether all of this discussion is related to frequency or substance use more generally.

Reviewer #2: The paper presented comprehensive information on the Substance use disorders and suicidality in youth. It is generally well written, and the topic is interesting. I only have some minor comments.

#1. The upper limit of the age was set 25 years old. There should be heterogeneity among several age groups, however, I couldn't find the results according to the age-level. The age-dependent meta-analytic results should be presented.

#2. Please describe in detail how the authors conducted the three-level meta-analytic model for continuous moderators. How is it different to the metaregression analysis?

6. PLOS authors have the option to publish the peer review history of their article (what does this mean?). If published, this will include your full peer review and any attached files.

Reviewer #1: No

Reviewer #2: No

---

## [Author Response · Author response to Decision Letter 0]

28 May 2021

JOURNAL/EDITOR

Comment 1. Please review your reference list to ensure that it is complete and correct. If you have cited papers that have been retracted, please include the rationale for doing so in the manuscript text, or remove these references and replace them with relevant current references. Any changes to the reference list should be mentioned in the rebuttal letter that accompanies your revised manuscript. If you need to cite a retracted article, indicate the article’s retracted status in the References list and also include a citation and full reference for the retraction notice.

Response: The reference list was reviewed for accuracy and no changes to the reference list were made.

Comment 2. Please ensure that your manuscript meets PLOS ONE's style requirements, including those for file naming. The PLOS ONE style templates can be found at

Response: The manuscript was checked for compliance with style requirements.

Comment 3. We note that this manuscript is a systematic review or meta-analysis; our author guidelines therefore require that you use PRISMA guidance to help improve reporting quality of this type of study. Please upload copies of the completed PRISMA checklist as Supporting Information with a file name “PRISMA checklist”.

Response: In our previous submission, the PRISMA checklist was included as Supporting information “S1_Checklist”. We changed the name of the file to “PRISMA checklist”.

REVIEWER 1

Overall, the manuscript by Rioux and colleagues will make an important contribution to the literature. The manuscript has multiple strengths including the following.

1) It examines direction in the relationship between SUD and suicide in the current literature and helps to summarize what is know on this important topic.

2) The study uses a well thought out methodology that that addresses potential biases in analyses. The authors present extensive analyses in detail to support their results.

3)The manuscript is well written and organized and includes discussion of important limitations. It also includes a comprehensive discussion next steps.

However, there are a few minor issues that should be addressed.

Comment 1. Overall, the discussion section would be improved if the paragraphs were broken up further. For instance, from page 19 line 291 to page 20 line 328 is one paragraph discussing moderators. This could be broken into several paragraphs. This issue is present throughout the discussion section, subsections (e.g., “limitations”) would help to navigate this discussion.

Response: We agree with the reviewer that the reader could be better guided through our discussion. Accordingly, we divided the discussion in subsections (i.e., after general objectives and results, we now have sections on the direction of association and age groups, moderators, limitations, clinical implications, and conclusion). As suggested, we also broke up longer paragraphs into shorter ones. The paragraph on moderators mentioned by the reviewer was broken up in three paragraphs based on general topics of moderators discussed (i.e., sex and gender, ethnic minority status, and other moderators, now lines 395-436 in track changes version, lines 322-361 in unmarked version). The information now under the “Direction of association and age groups”, “Limitations” and “Clinical implications” sections were all one long paragraph, which was broken into two paragraphs for each section.

Comment 2. The discussion of substance abuse frequency is not entirely clear. Lines 300-305 discusses this issue. Although the opening sentence indicates discussion of substance use frequency, the later lines use the term “substance use” instead. It is not clear that whether all of this discussion is related to frequency or substance use more generally.

Response: We clarified this sentence, which refers to studies on substance use frequency (the term “substance use” was changed for “substance use frequency”, line 381 in track changes version, line 332 in unmarked version).

REVIEWER 2

The paper presented comprehensive information on the Substance use disorders and suicidality in youth. It is generally well written, and the topic is interesting. I only have some minor comments.

Comment 1. The upper limit of the age was set 25 years old. There should be heterogeneity among several age groups, however, I couldn't find the results according to the age-level. The age-dependent meta-analytic results should be presented.

Response: We agree with the reviewer that effects by age are important to examine, but we could not include them due to the variety in average ages, age ranges, and follow-up lengths of the studies reviewed. This was mentioned in the discussion, but it was one sentence in the middle of a long paragraph and we now realize it could easily be missed. Accordingly, we made the sentence mentioning this clearer (lines 331-332 in tracked changes version, 284-285 in unmarked version) and the information is easier to find as it is now at the beginning of the section titled “Direction of association and age groups.

Comment 2. Please describe in detail how the authors conducted the three-level meta-analytic model for continuous moderators. How is it different to the metaregression analysis?

Response: The three-level meta-analyses uses a regression model. We added a sentence explaining that the analyses are equivalent to a meta-regression, but that they control for clustering of effect sizes and allow examining moderators that vary within and/or between samples (lines 165-168 in track changes version, 161-164 in unmarked version).

---

## [Decision Letter · Decision Letter 1]

26 Jul 2021

Substance use disorders and suicidality in youth: A systematic review and meta-analysis with a focus on the direction of the association

PONE-D-20-38935R1

Dear Dr. Castellanos-Ryan,

We’re pleased to inform you that your manuscript has been judged scientifically suitable for publication and will be formally accepted for publication once it meets all outstanding technical requirements.

Kind regards,

Kyoung-Sae Na, M.D.

Academic Editor

PLOS ONE

Additional Editor Comments (optional):

Reviewers' comments:

Reviewer's Responses to Questions

**Comments to the Author**

1. If the authors have adequately addressed your comments raised in a previous round of review and you feel that this manuscript is now acceptable for publication, you may indicate that here to bypass the “Comments to the Author” section, enter your conflict of interest statement in the “Confidential to Editor” section, and submit your "Accept" recommendation.

Reviewer #2: All comments have been addressed

2. Is the manuscript technically sound, and do the data support the conclusions?

Reviewer #2: Yes

3. Has the statistical analysis been performed appropriately and rigorously? 

Reviewer #2: Yes

4. Have the authors made all data underlying the findings in their manuscript fully available?

Reviewer #2: Yes

5. Is the manuscript presented in an intelligible fashion and written in standard English?

Reviewer #2: Yes

6. Review Comments to the Author

Reviewer #2: (No Response)

7. PLOS authors have the option to publish the peer review history of their article (what does this mean?). If published, this will include your full peer review and any attached files.

Reviewer #2: No

---

## [Editor Report · Acceptance letter]

28 Jul 2021

PONE-D-20-38935R1 

Substance use disorders and suicidality in youth: A systematic review and meta-analysis with a focus on the direction of the association 

Dear Dr. Castellanos-Ryan:

I'm pleased to inform you that your manuscript has been deemed suitable for publication in PLOS ONE. Congratulations! Your manuscript is now with our production department. 

Kind regards, 

on behalf of

Dr. Kyoung-Sae Na 

Academic Editor

PLOS ONE